# COVID-19: Evidenced Health Disparity

**Ayodeji Iyanda** [1,*] **[ID], Kwadwo Boakye** [2] **and Yongmei Lu** [1] **[ID]**

1   Department of Geography, Texas State University, San Marcos, TX 78666, USA; yl10@txstate.edu
2   School of Biological and Population Health Sciences, Oregon State University, Corvallis, OR 97331, USA; boakyek@oregonstate.edu
*   Correspondence: aei11@txstate.edu

**Definition:** Health disparity is an unacceptable, unjust, or inequitable difference in health outcomes among different groups of people that affects access to optimal health care, as well as deterring it. Health disparity adversely affects disadvantaged subpopulations due to a higher incidence and prevalence of a particular disease or ill health. Existing health disparity determines whether a disease outbreak such as coronavirus disease 2019, caused by the severe acute respiratory syndrome coronavirus 2 (SARS-CoV-2), will significantly impact a group or a region. Hence, health disparity assessment has become one of the focuses of many agencies, public health practitioners, and other social scientists. Successful elimination of health disparity at all levels requires pragmatic approaches through an intersectionality framework and robust data science.

**Keywords:** health disparity; intersectionality; data science; machine learning; public health; GIS

## 1. Introduction

The coronavirus disease 2019 (COVID-19) outbreak caused by severe acute respiratory syndrome coronavirus 2 (SARS-CoV-2) was declared a pandemic by the World Health Organization on 11 March 2020 due to its growing prevalence, particularly its increase in morbidity and mortality over a wide geographic area [1], and has since generated considerable attention. The infection rate varies across geographic areas, indicating spatial and regional disparities. Evidence of regional and within-country variations in coronavirus infection and mortality has been well documented [1–5]. Shortly after the first outbreak in Wuhan, China, several other American and European countries exhibited clusters of COVID-19 infections. Not long after, the United States became a significant hotspot of COVID-19 morbidity and mortality. Figure 1 highlights the 7-day average cumulative cases of COVID-19 in five countries: the United States, Canada, Germany, the Netherlands, and Nigeria. Recently, when the distribution of COVID-19 vaccination became more accessible in most communities in the United States, the daily hospitalization cases due to infection and related death decreased significantly [6–9]. However, overwhelming evidence indicates differences in infection and associated death rates across different populations [2,3].

The recent COVID-19 pandemic revealed many things ranging from the possibilities of remote learning/working, virtual recruitment, and service delivery such as telemedicine. The pandemic has further shown additional evidence of health disparity in our society, particularly in health outcomes and access to health care systems. As a background for this paper, Figure 2 shows the intersection between structural inequality as the root cause of health disparity, public health, and data science. We believe that the adoption of intersectionality and data science approaches would help foster data-driven interventions in addressing health disparities.

Health disparity is an important theme that has generated much interest from different domains as well as a significant public health debate that must not be overlooked in any society and in chronicling the health outcomes of COVID-19, particularly in the United States. This entry paper examines the concept of health disparity within the

intersection of public policy, public health, and data science (Figure 2). Furthermore, this contribution explores the definition of health disparity and determinants of health regarding COVID-19 from different countries, with specific emphasis on high-income countries with diverse populations such as the United States and the United Kingdom. For example, the United States has the most extensive published papers on health disparity, mainly from racial and ethnic, social class, gender, and sexual orientation angles, and serves as a reference point for research on racial and health disparity.

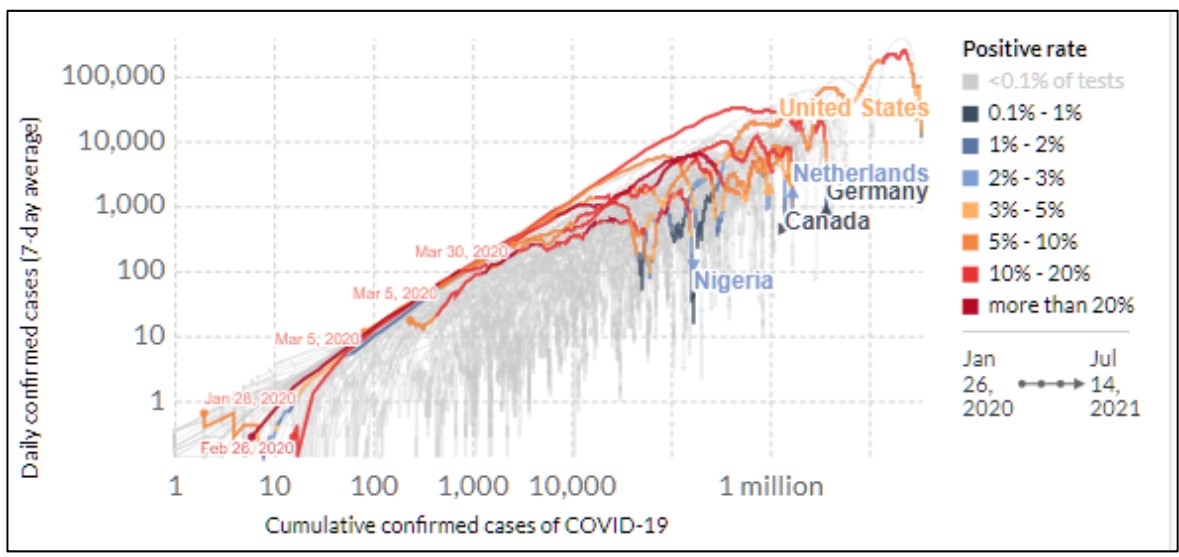

**Figure 1.** Cumulative confirmed global cases of COVID-19 (data source: www.Ourworldindata.org accessed 13 July 2021, Ritchie et al. [10]).

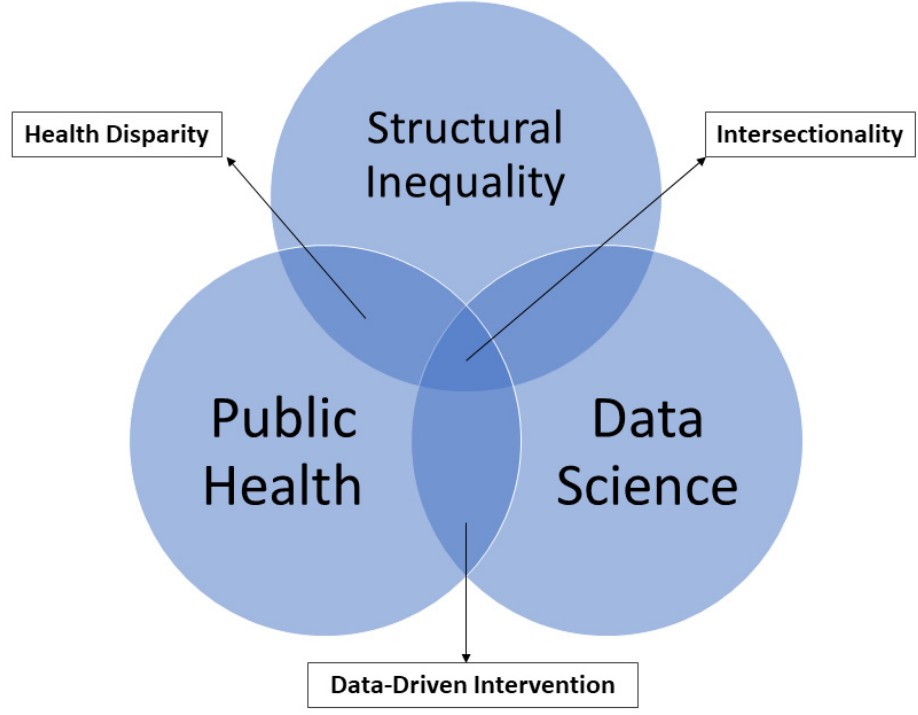

**Figure 2.** The intersection between structural inequality, public health, and data science.

In addition, this paper provides a general overview of the implication(s) of health disparities for achieving viable public health policies. Considering different approaches to measuring health disparity, this study presents how the field of public health and other

allied (health) disciplines could leverage the power of data science such as geographic information science/systems (GIS), machine learning, and artificial intelligence to help solve some of the societal issues including the current COVID-19 pandemic. In addition, this paper contributes to the field of spatial epidemiology and public health by showing how data science and intersectionality frameworks can be fused to improve general/public health. Finally, this paper concludes with some expert-based recommendations toward reducing health disparity.

## 2. Defining Health Disparity

Health disparity is a ubiquitous phrase in most health writeups found on the pages of printed and digital media, journals, and reports from governmental and non-governmental organizations, as well as in discussions among health professionals, policymakers, and the general public. However, the conceptualization and determination of health disparity vary by geographic location. Table 1 presents the various but similar definitions of health disparity and health inequality as used within and outside the United States. For instance, in the United States, health disparity determines whether health, broadly used, differs by comparing different health outcomes among various population groups and subpopulations of interest. In contrast, the terms "health inequality" and "health inequity" are frequently used outside the United States. For example, the World Health Organization consistently uses the term "inequity," which connotes the differences in health that are unnecessary and avoidable but unfair and unjust [11].

Generally, health disparity refers to a higher burden of illness, injury, disability, or mortality experienced by one group relative to another. Determining which health situations could be considered unfair or unjust depends on the place and time of examination. However, the widely used measure hinges on the degree of choice involved. Essentially, any strive toward attaining equity and health is "not to eliminate all health differences so that everyone has the same level and quality of health, but to reduce or eliminate those which result from factors considered to be both avoidable and unfair" [12]. This prompts the question, what is avoidable and/or unavoidable health inequality? However, the answer to this question is beyond the scope of this article (for additional readings, see: Braveman [11,13]; also Whitehead [12]).

The National Institute of Minority Health Disparities' (NIMHD) definition [14] of health disparity highlights health differences among different populations, emphasizing disadvantaged populations. According to the legislation that birthed the NIMHD, what defines health disparity includes several factors characterized by poorer health outcomes, indicated by the overall disease incidence, prevalence, morbidity, mortality, or survival in the population compared with the general population. The NIMHD recognizes health disparity populations as racial/ethnic minorities, socioeconomically disadvantaged populations, underserved rural populations, and sexual and gender minorities.

Similarly, the United Kingdom's National Health Service describes adversely affected populations within the "Inclusion Health" framework. Inclusion health refers to "a number of groups of people who are not usually well provided for by health care services and have poorer access, experiences and health outcomes. The definition covers people who are homeless and rough sleepers, vulnerable migrants (e.g., refugees and asylum seekers), sex workers, and those from the Gypsy, Roma and Traveller communities" [15]. Nevertheless, inclusion health or each of the compositions of people affected by health disparity could be examined in more detail based on how they have been impacted by COVID-19 if and when detailed health data are available. The related findings could help inform better intervention policies to reduce health disparity among different population groups.

**Table 1.** Different definitions of health disparity and health inequality.

| Organization, Country | Definition |
|---|---|
| National Institute of Minority Health Disparities [14] | A health difference, determined on the basis of one or more health outcomes that adversely affect disadvantaged populations. |
| Healthy People 2020 [16] | A particular type of health difference closely linked with social, economic, and environmental disadvantages. |
| Centers for Disease Control and Prevention [17] | Preventable differences in the burden of disease, injury, violence, or opportunities to achieve optimal health experienced by socially disadvantaged racial, ethnic, and other population groups and communities. |
| Institute of Medicine [18] | A health service disparity between population groups is determined as differences in treatment or access not justified by differences in the health status or preferences of the groups. |
| National Health Service [15] | Unfair and avoidable differences in health across the population and between different groups within society. |

Compared to the NIMHD's definition of health disparity that centers on disadvantaged populations, Healthy People 2020's definition is more expansive. Healthy People 2020's definition of health disparities underscores social, economic, and environmental resources differentiation. Hence, health disparities adversely affect groups of people who have systematically experienced greater obstacles to health based on their racial or ethnic group, religion, socioeconomic status, gender, age, mental health, cognitive, sensory, or physical disability, sexual orientation or gender identity, geographic location, or other characteristics historically linked to discrimination or exclusion [19]. Similarly, England's National Health Service defines health inequality as unfair and avoidable differences in health across the population and between different groups within society (Table 1). Carter-Porkras and Baquet [20] presented other definitions of health disparity within the United States context. For example, the Taskforce on Black and Minority Health, Department of Health and Human Services, in 1985, used excess deaths (excess death is referred to as the difference between the number of deaths observed in minority populations and the number of deaths which would have been expected if the minority population had the same age- and sex-specific death rate as the non-minority population) of six key health indicators. These health indicators are cancer, cardiovascular disease and stroke, cirrhosis, diabetes, homicide and unintentional injuries, and infant mortality.

The Institute of Medicine differentiates health disparity between minority and non-minority groups based on clinical appropriateness, needs, and patient preferences; discrimination at the individual patient–provider level that results from biases, prejudices, stereotyping, and uncertainty in clinical communication and decision making; and, lastly, the operation of health care system functions. The American Medical Association (AMA) also recognizes health disparities in health care systems.

In 2000, the United States Department of Health and Human Services launched Healthy People 2010, which had two goals: (1) to improve the overall health status of Americans and (2) to eliminate racial and ethnic health care disparities. The AMA was deemed fit to promote this agenda. In 2004, the association established the Commission to End Health Care Disparities, while the National Medical Association was established to address gaps in health care. As part of this agenda to eliminate racial and ethnic health care disparity, the AMA strives to promote diversity by increasing the share of physicians through its policies and advocacy work and therefore proposed immigration reform as a means to eliminate health care disparities, prioritized access to care for patients with disabilities, and proposed its intent to tackle unequal treatment [21]. Hence, closing health disparities requires an intersectional approach and bold and unapologetic political willingness.

## 3. Intersectionality Framework and Health Disparity

Although several studies [22–25] have underscored the importance of intersectionality in adverse health outcomes, such as the current COVID-19 pandemic, its application

is scarce. The adoption of intersectionality in health analysis, intervention, and policy could reduce health disparity. According to Kimberly Crenshaw [26], "intersectionality is a lens through which one can see where power comes and collides, where it locks and intersect." Intersectionality is accepting that different people have unique experiences of discrimination and privilege at different stages in their lifetime. Scholars have employed an intersectionality framework to examine the experiences of individuals belonging to different marginalized groups. Some of the indicators that should be considered in order to achieve intersectionality include race, ethnicity, gender identity, class, languages, skin color, neurodiversity, religions, (dis)ability, sexuality, mental health, age, ecological systems (neighborhood, school, office, rural–urban, local, and federal policies), education, citizenship and immigration status, housing, and body size (this list could go on). The intersectionality framework could be helpful in transformative health disparity research, especially in understanding disparity in COVID-19 health outcomes.

Furthermore, the intersectionality framework posits that inequality is experienced through multiple contexts, and it should not be examined from a single standpoint. For example, the intersectionality framework could help us understand why Black women are more likely to die through childbirth than White women (maternal mortality disparity) or why there is a high rate of incarceration or racially profiled crimes among Black men or Black adolescents/youth. Within the current COVID-19 pandemic context, we could continue to ask questions about whether more older Black, Hispanic, and Asian women were infected or died of COVID-19 and why? These are the questions intersectionality will help shed some light on if applied.

Using the intersectionality approach, Sekalala et al. [24] analyzed the intersection between age and sex to determine vaccination prioritization based on human rights needs. In addition, Eaves and Falconer Al-Hindi [23], in a short article published in Dialogues in Human Geography, claimed that "geographical research that is designed, conducted, and analyzed within an intersectional framework generates better geographical scholarship". Hence, researchers should consider how the concept of intersectionality could reveal more disparity by intersecting some of these indicators. Unfortunately, as seen in Table 2, this concept has received limited attention in the study of COVID-19.

### 3.1. COVID-19 Disparity

Regarding disparity in COVID-19 outcomes, it is evident that the disease has exacerbated health inequality. The health impacts of COVID-19 are unevenly distributed, particularly for underrepresented racial and ethnic minorities and migrants [27–29]. Figure 3 shows the pattern and distribution of COVID-19 deaths based on the 2017 population density per square kilometer for some selected countries. It is worth emphasizing that the pandemic did not cause health disparity in that some minority ethnic groups died at a significantly higher proportion; it only shed more light on the longstanding health disparity in various countries [30–33]. Existing data on infections, hospitalizations, and deaths reveal significant variation among and within regions and communities [27,28,34–36], prompting questions about health inequality and disparity; they have also stirred curiosity to know which populations are at higher risk and why the risk is higher for one group than others [27]. The within-country health determinants may explain this variation among the countries of the world. This type of question is what proponents of social, health and environmental justice have been pursuing before the outbreak of the COVID-19 pandemic.

**Table 2.** Common variable of interest, intersectionality framework, and data analytical techniques in COVID-19 research.

| Study | Location | Health Outcomes | Intersectionality Framework | Common Variables of Interest | | | | | | | | | Data Analytical Techniques | | |
|---|---|---|---|---|---|---|---|---|---|---|---|---|---|---|---|
| | | | | Violence | Health Care | Public Policies | Mobility | Racial/Ethnic Heterogeneity | Sociodemographic | Environmental Injustice | Health Disparity | Geographic/Temporal Disparity | GIS/Spatial Statistics | AI, Machine, and Deep Learning | Aspatial |
| Chaudhuri et al., 2021 | UK | Age-adjusted COVID-19 morality | | | | | | x | x | | x | x | SAR | | OLS |
| Iyanda et al., 2020 | Global | COVID-19 outbreak | | | x | x | | | x | | x | x | MGWR | | OLS |
| Iyanda et al., 2021 | USA | COVID-19 case fatal ratio | | | | | | x | x | | x | x | GWR | | Poisson |
| Louis-Jean et al., 2020 | USA | COVID-19 | | | | | | x | x | | | | | | |
| Allen Et al., 2020 | USA | COVID-19 confirmed cases, deaths | | | | | | x | x | x | x | x | Thematic mapping | | OLS |
| Abedi et al., 2020 | USA | COVID-19 infection rate | | | | | x | x | x | | x | x | Map overlay | | OLS, Pearson correlation, Forest Plot |
| Chen et al., 2020 | China | COVID-19 Mortality rate | | | | | | | | x | | | Remote sensing | | Difference-in-Difference |
| Adams, 2020 | Canada | COVID-19 | | | | x | x | | | x | | x | | | Polynomial regression |
| Lippi et al., 2020 | Italy | COVID-19 infection | | | | | | | | x | | x | | | Pearson's correlation |
| Berman & Ebisu, 2020 | USA | COVID-19 infection | | | | x | | | | x | | x | | | *t*-test |
| Travaglio et al., 2021 | UK/England | COVID-19 mortality | | | | | | | | x | x | x | Heatmap | | GLM, BLR |
| Arimiyaw et al., 2020 | SSA region | COVID-19 | | | | X | | | | x | | | | | |
| Fattorini & Regoli, 2020 | Italy | COVID-19 | | | | | | | | x | x | x | Thematic mapping | | Pearson's correlation |
| Bashir et al., 2020 | Germany | COVID-19 confirmed cases, deaths, recoveries | | | | | | | | x | x | | | | wavelet transform coherence; correlation |
| Bashir et al., 2020 | California, USA | COVID-19 confirmed cases, deaths | | | | | | | | x | x | x | Thematic mapping | | Spearman's/Kendall correlation |
| Chakraborty, 2021 | USA | COVID-19 incidence rate | | | | | | x | x | x | x | x | LISA | | OLS, GEE |
| Terrel & James, 2020 | Louisiana, USA | COVID-19 deaths | | | | | | x | x | x | x | x | | | Spearman correlation; Shapiro-Wilks's test |
| Martinez Dy & Jayawarna, 2020 | UK | COVID-19 impacts | | | | | | x | x | | | | | | |
| Krzysztofowicz & Osińska-Skotak, 2021 | Poland | COVID-19 Vaccination | | | | | | | | | | | Thiessen Polygon | | |

**Table 2.** *Cont.*

| Study | Location | Health Outcomes | Intersectionality Framework | Violence | Health Care | Public Policies | Mobility | Racial/Ethnic Heterogeneity | Sociodemographic | Environmental Injustice | Health Disparity | Geographic/Temporal Disparity | GIS/Spatial Statistics | AI, Machine, and Deep Learning | Aspatial |
|---|---|---|---|---|---|---|---|---|---|---|---|---|---|---|---|
| Bachtiger et al., 2020 | Generalized | COVID-19 | | | x | | | | | | | | | x | |
| Punn et al., 2020 | Global | COVID-19 confirmed cases, death, recovery | | | | | | | | | | | | x | |
| Cavaljal et al., 2018 | Philippines | Dengue | | | | | | | | | | | | x | |
| Alimadadi et al., 2020 | Generalized | COVID-19 | | | | | | | | | | | | x | |
| Kushwaha et al., 2020 | | COVID-19 | | | | | | | | | | x | | x | |
| Pinter et al., 2020 | Hungary | COVID-19 | | | | | | | | | | | | x | |
| Li et al., 2021 | Multi-country | | | | | | | x | x | | | x | | x | |
| Kuo & Fu, 2021 | USA | COVID-19 infection | | | | | | | | | | | | x | |
| Mollalo et al., 2018 | Iran | Sandfly; Cutaneous leishmaniasis | | | | | | | | | | | x | x | Pearson's correlation |
| Mele & Magazzino, 2020 | India | COVID-19 | | | | | | | | | | | | x | |
| Biana, 2020 | Philippines | COVID-19 | | x | | x | | | x | | x | x | | | |
| Sonu et al., 2021 | USA | COVID-19 | x | x | | | | | | | x | | | | |
| Wilson et al., 2020 | USA | COVID-19 | | x | | | | x | | x | x | | | | |
| Reicher & Stott, 2020 | UK, USA, France | COVID-19 | | x | | | | | | | x | | | | |
| Corpuz 2021 | Philippines | COVID-19 | | x | | x | | | | | x | | | | |
| Njoku et al., 2021 | USA | COVID-19 | | x | | x | | x | | | x | | | | |
| Joseph-Salisbury et al., 2021 | UK | COVID-19 | | x | | x | | x | | | | | | | |
| Gibson et al., 2021 | USA | COVID-19 | | x | | x | | x | | | | | | | |
| Coyne & Yatsyshina, 2020 | Generalized | COVID-19 | | x | | x | | | | | | | | | |
| Bailey et al., 2020 | Generalized | COVID-19 | | x | | x | | x | | | x | | | | |
| Elias et al., 2021 | Generalized | | | | | | | | | | | | | | |

OLS ordinary least squares; GEE general estimating equation; BLR binomial linear regression; LISA local indication of spatial autocorrelation/association; SAR spatial autoregressive; GWR geographically weighted regression; MGWR multiscale GWR; GIS geographic information systems; AI artificial intelligence.

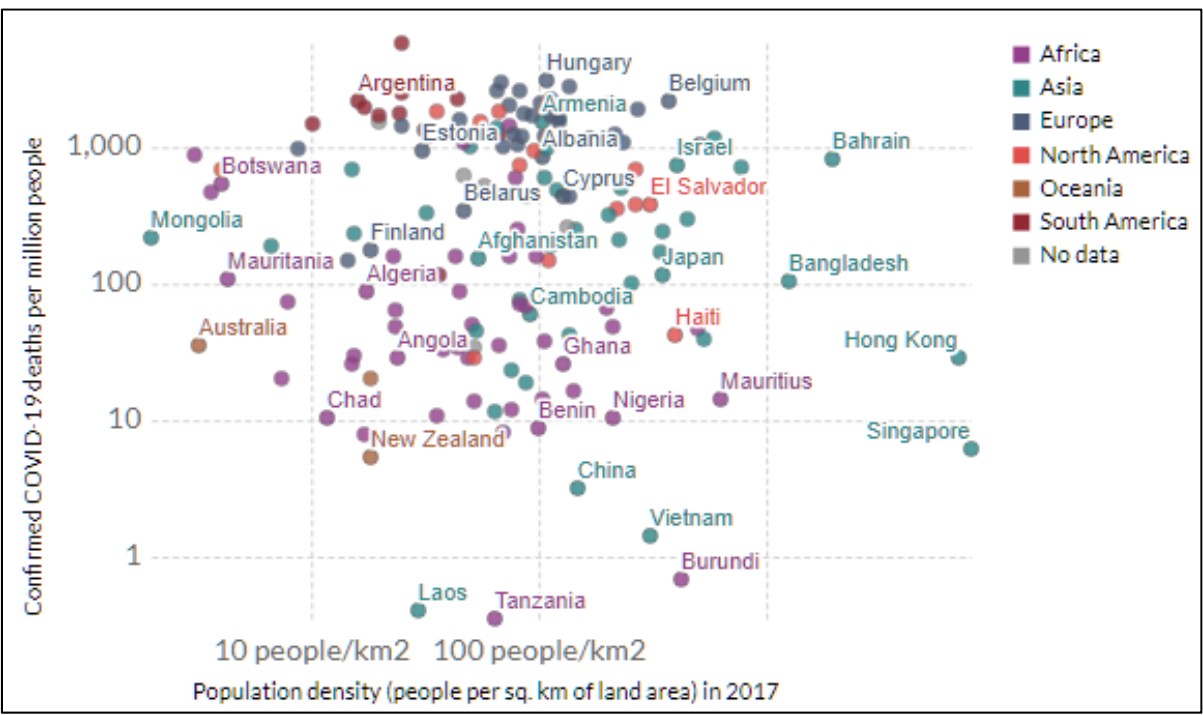

**Figure 3.** COVID-19 deaths versus population density based on data from 22 January to 14 July; time 20:03 London time. (data source: www.ourworldindata.org/Coronavirus-death accessed 13 July 2021; Ritchie et al., [10]). Note: countries with a < 1 million population are excluded from the visualization.

### 3.2. COVID-19 and Environmental Injustice

The advent of COVID-19 adds "icing on top of the cake" of environmental justice research that has been in progress for several decades, particularly in the United States. For example, in a roundtable discussion [37], Robert Billard referred to the COVID-19 pandemic as "a heat-seeking missile that is targeting the most vulnerable populations, and the bull's-eye is actually the environmental justice communities, the communities that are the poorest, that are the most polluted, that are the sickest when it comes to comorbidity, and the result of this heat-seeking missile called COVID-19 is a death bomb." In addition, many studies situate research on COVID-19 health disparity within the broader framework of environmental injustice [38–48]. These studies examine environmental variables such as air pollution in different geographic locations from countries in North America through Africa to Europe (e.g., Germany, Italy, and Britain).

Several studies allude to the reduction in air pollutants (e.g., $PM_{2.5}$, $NO_2$, $O_3$) from the reduced human mobility due to lockdowns [38,40,49], and several others indicate a correlation between the COVID-19 burden and environmental pollution [50,51], particularly in economically disadvantaged communities and areas with a high concentration of people of color [35,47,52–56]. For example, Chakraborty [54], using the local indicator of spatial association, showed convergence between the COVID-19 incidence rate and the respiratory risk from hazardous air pollutants across counties in the United States. Additionally, his study indicated that being Black and living in socioeconomically deprived neighborhoods increased the COVID-19 incidence, supporting a similar study from the United Kingdom [34].

Mackey et al.'s [28] systematic review of 37 cohort and cross-sectional studies and 15 ecological studies, for example, showed that African American/Black and Hispanic populations experience disproportionately higher rates of COVID-19 infection, hospitalization, and coronavirus disease-related mortality compared with non-Hispanic White populations, but not higher case fatality rates. In addition, the Atlantic, a COVID-19 dashboard that

tracks infection and deaths by race/ethnicity across the United States, shows evidence of health disparity [57]. For example, Alabama, where 27% of the population is Black or African American, had a higher infection rate (29%) and the number of deaths (31%) for its Black people compared to its White population.

Chaudhuri et al.'s [34] work on the interaction of ethnicity and socioeconomic deprivation on COVID-19 mortality in the United Kingdom showed that Black, Asian, and Minority Ethnic (BAME) populations are at higher risk of developing COVID-19 and likely to experience severe health-related outcomes compared to White British populations. Relying on disaggregated ethnicity data, the England study [34] reported that the age-adjusted COVID-19 mortality rate was higher in minority neighborhoods in the highest deprivation quartile, with a high concentration of Black-African (regression coefficient, β 2.86; 95% CI 1.08–4.64), Black-Caribbean (β 9.66: 95% CI 5.25–14.06), and Bangladeshi (β 1.95: 95% CI 1.14–2.76) populations.

Using a combination of statistical and geographic information system techniques, Iyanda and colleagues [1] used 356 days of COVID-19 cases and deaths data to examine disease fatality in the United States (Table 2). They showed that African American/Black and Hispanic populations experienced disproportionately higher case fatality rates in the rural counties with a higher share of minority populations. Their findings corroborate earlier data by the Centers for Disease Control and Prevention that show that Black and Hispanic communities are three and two times more likely to be infected and die from COVID-19, and other sources have shown similar racial/ethnic disparities [35,36,53,54,57,58]. These results reinforce how structural racism may have shaped the distribution of social determinants of health and promotes social risk factors. More interestingly, action within the health care system has been hampered by a lack of understanding of how to abate the impact of institutional and structural racism on influencing minority health [59].

### 3.3. COVID-19 and Incidence of Violence

Table 2 presents ten studies examining different forms of violence (e.g., police brutality, riots, shooting), mostly during COVID-19. In addition to the institutional violence, environmental injustice, and segregationist and discriminatory laws that faced, majorly, the minorities and immigrants in most countries is the COVID-19 pandemic which stretches structural racism in many countries [30,37,60–65]. It has become common knowledge that "pandemics always follow the fault line of society" [33]—structural or institutional racism. Research indicates that racism and xenophobia increased during the coronavirus outbreak due to the increasing nationalism and populism [30]. For example, Gibson et al. [66] chronicled institutional responses to protests and outbreaks of violence due to the deaths of several African Americans, including Ahmaud Arbery, Breonna Taylor, and George Floyd, amidst the global COVID-19 pandemic in 2020. Similar to the experience of the violence of the #BlackLivesMatter movement in the United States, there were violence outbreaks in the United Kingdom due to longtime racism. Within the Philippine context, Corpuz [62] called to end the police brutality of Filipinos, which was exacerbated during the coronavirus outbreak. Biana [67] argued that the outcome in health disparity could be related to a matter of class. For example, being Black in the United States is considered a risk factor for disease vulnerability, while being poor in the Philippines means "to be neglected and be reduced to nothing." This assertion might as well explain the excess infections and deaths from COVID-19 in different geographies.

### 4. Determinants of Health Disparities

According to Healthy People 2020, several powerful and complex relationships exist between biology or genetic compositions, individual behavior, socioeconomic status, the physical environment, discrimination, racism, literacy levels [16], and non-pharmaceutical policies that influence the susceptibility, exposure, transmission, infection, recovery, and fatality rate of COVID-19 and the probability of receiving timely as well as adequate health care.

Common health determinant factors in developed and developing countries include the availability of or lack of access to high-quality education, nutritious food, decent and

safe housing, affordable and dependable public transportation, culturally and socially sensitive health care providers, health insurance, and clean water and air. Several studies have examined how the COVID-19 pandemic affects some of these factors, including air quality, business, nutrition, and health care services [68–71]. In addition, available data indicate that a more significant proportion of racial and ethnic minorities depend on government assistance programs such as SNAP. Moreover, most of them have no health insurance, a critical barrier to access adequate health care, particularly during a pandemic. Research such as the work of Chakraborty [54] in the United States and Iyanda et al. [1] global ecological study highlight the importance of health insurance during the COVID-19 pandemic. Thus, it is not surprising that COVID-19 only shows the evidenced health disparity that has existed for decades in most countries.

### 4.1. Social and Structural Determinants of Health

Social determinants of health are the non-medical factors that influence health outcomes, and they are generally the conditions in which people are born, grow, work, live, and age, and the broader set of forces and systems shaping the conditions of daily life [72]. COVID-19 has exhibited various characteristics based on some of the previously highlighted conditions. As evidenced by the existing data, the disparities in COVID-19 morbidity and mortality in the United States were deepened mainly by structural factors such as racism [73–77]. There are different nomenclatures of structural racism. Online dictionaries describe it as societal racism, state racism, covert racism, systemic racism, and institutional racism. Irrespective of the adopted nomenclature of racism, it has significant implications on health and health outcomes.

Structural, institutional, and systemic racism synonymously refer to structures with procedures or processes that disadvantage people of color, particularly African Americans. An excellent example of structural racism is the "redlining" policies that discriminated against Black communities regarding the opportunity to secure loans [78]. Since housing is a critical determinant of health, discriminatory housing policies play a critical role in the health conditions of the group of people being discriminated against. For example, breast, colorectal, cervical, and lung cancers, as well as birth outcomes and asthma, have been shown to correlate with redlining policy [79–84]. In addition, housing congestion and overcrowding are major underlying factors for the diffusion of infectious diseases such as COVID-19. Therefore, the authors adopt structural racism in this article, which is more prevalent and favored in health research.

### 4.2. Structural Racism

Structural racism is how societies foster discrimination through mutually reinforcing inequitable systems. It is usually embedded through laws within society or different levels of an organization. For example, during the peak of the COVID-19 outbreak in the United States in 2020, more racial and ethnic minorities were infected and died more than their White counterparts. COVID-19 impacted more Blacks or African Americans and Hispanics than any other racial and ethnic group in the United States. Studies on COVID-19 focusing on geographic disparity in the infection, death, and case fatality rates show that Black/African Americans living in inner cities with a high population density were at high risk [85]. In addition, research on the rural–urban disparity of COVID-19 case fatality ratio in the United States indicates that Blacks in rural counties were disproportionately burdened [27]. These disparities are inherently linked with racial and housing segregation, intergenerational poverty, and structural racism [74].

Historically, African Americans had the least access to health care in the United States due to their inability to access health insurance, mainly because of the low income and low employment rate among this subpopulation [73,75,76]. In addition, African Americans were more impacted because they have limited access to single-family homes due to historical institutional discriminatory policies (e.g., redlining and FHA policies) that force them to converge in mostly the poorest, densely populated inner-city spaces [78]. Similar experiences exist for Black, Asian, and Minority Ethnic (BAME) groups in the

United Kingdom [34,86–89] and indigenous populations around the world [90–93]. Generally, minorities are disproportionately overrepresented in neighborhoods of concentrated poverty [94], which is detrimental to mental and physical health and wellbeing. Consequently, during the lockdown, the recommendations to maintain social distancing were less realistic for them and their communities.

*4.3. Age and Gender Disparity*

Globally, senior adults aged 65 and above are more likely to die from COVID-19 complications than other age groups, and more males than females were infected and died from the disease. What research has not readily shown is age disparity in minority groups. In addition, the observed sex and gender differences in COVID-19-related health outcomes could be explained by the sex- and gender-based medicine (SGBM) model that incorporates how biological sex and the socio-cultural aspects of gender affect health and illness [95]. Previous epidemiological research on similar respiratory infectious diseases such as the 2003 severe acute respiratory syndrome (SARS) and the 2012 Middle East respiratory syndrome (MERS) outbreaks show evidence of sex and gender disparities. Age-based research on COVID-19 also demonstrates age group disparity of COVID-19 infection and death rates within the United States context, and several examples are available globally. Sadly, despite the United States having some of the highest coronavirus cases and deaths, comprehensive age- and sex-specific data concerning patients in the United States are not currently available to the community of researchers to effectively inform health policy.

*4.4. Case Study of COVID-19*

COVID-19, as a peculiar case study, could help us better understand racism, structural racism, inter-racism, and equity. A deep understanding of equity matters because it helps us better understand how resources are distributed according to the underlying structure. For example, in the case of COVID-19, we could go back and analyze the distribution of PPE, hospital beds, access to care and services, and, now, the distribution of COVID-19 vaccines by race and ethnicity, age groups, and gender and sexual orientation. We can then ask whether minority communities that are most impacted receive the required care and can access vaccines as needed. Existing data on vaccine distribution at the county level indicate that counties identified as the most vulnerable and most impacted received fewer vaccines [96], indicating the existing health disparity by geography, social class, and racial and ethnic groups.

**5. Implication for Achieving Global and Sustainable Health**

The variances in the definition of health disparity have significant policy implications in obtaining the desired health outcomes, particularly for reaching global health goals such as the United Nations' Sustainable Development Goals and other multilateral agencies (e.g., World Health Organization and World Bank). Figure 4 presents a snapshot of global disparity in the distribution of coronavirus vaccination. A stark difference exists between developed and less developed regions. For example, the dose per 100 people varies between 4.2 in Africa and 78 in North America. Most of the prominent multilateral organizations are all part of the United Nations, intending to improve the health and wellbeing of all global citizens. Since its creation in 1945, the United Nations has provided a common platform where all the world's nations can gather together, discuss common problems such as historical pandemics and epidemics, and find shared solutions that benefit humanity.

Although some multilateral agencies do not directly seek to improve global health, they are essential mechanisms to promote healthy living. A good reference, in this case, is the World Bank, whose goal is to alleviate poverty at all levels by providing loans, credits, and grants to developing/emerging countries toward improving the standard of living of citizens in developing nations. Additional goals include implementing various development projects such as education, health care, agriculture, environmental and natural resource management, infrastructure, and other relevant projects toward general improved wellbeing. For example, at the onset of the COVID-19 pandemic, the World Bank

highlighted 34 health indicators referred to as the "World Bank Indicators of Interest to the COVID-19 Outbreak," partly to investigate the global disparities and drivers of COVID-19 (see [1]). Despite the fact that the United States is an active player and frontline contributor in helping to fight global health issues and other socioeconomic matters, the COVID-19 pandemic hit hard on the "World's Police" country [97] and makes it the global hotspot of COVID-19 infection and mortality.

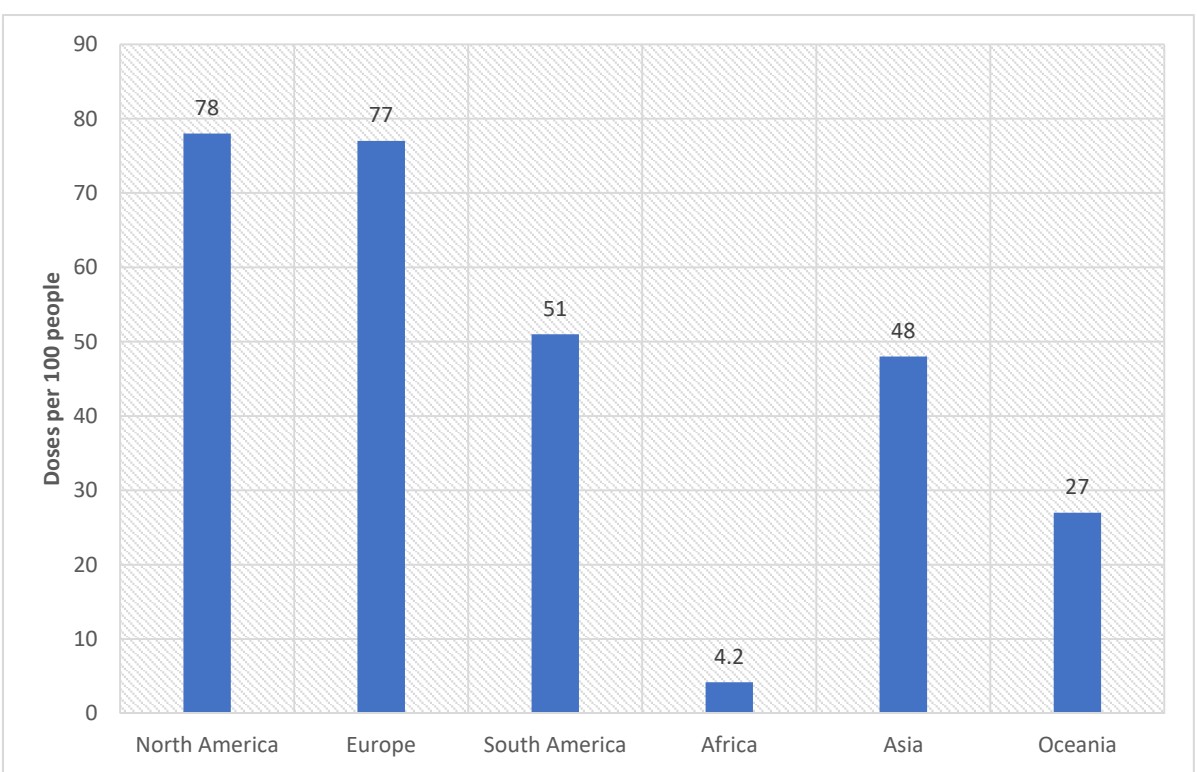

**Figure 4.** The global disparity in the COVID-19 vaccination doses per 100 people (data source: The New York Times [98]).

The United States Centers for Disease Control and Prevention (CDC) developed a social vulnerability index (SVI) for determining the impact of the COVID-19 community infection rate and morbidity and mortality rates. Briefly, social vulnerability refers to the potential adverse effects on communities caused by external stress on human health, including natural/human-caused disasters and disease outbreaks. Therefore, a pragmatic effort towards reducing social vulnerability can have a longtime effect on decreasing human exposure, suffering, and economic loss. Based on the CDC's SVI, counties in the United States were categorized into low to high social vulnerability [96].

Typically, neighborhoods in the highest social vulnerability tertile should be prioritized for distributing COVID-19 vaccines to quickly curtail the excess COVID-19-related morbidity and mortality, which will eventually address two common health issues—equality and equity. To address inequality, a similar allocation of health resources such as the supply of COVID-19 vaccines should be supplied proportionately based on the population across different jurisdictions or health regions. Likewise, the issue of equity can be addressed by prioritizing communities most affected by the pandemic. Nevertheless, the two issues of equity and inequality should be judiciously managed to avoid inter-racial and political conflicts.

## 6. Measuring Health Disparity

To monitor trends in health disparities in federally funded programs, Section 4302 of the Patient Protection and Affordable Care Act (ACA) suggests the need to improve data collection to reveal any evidence of health disparity [99]. However, a comprehensive and accurate measure of health disparity has become a significant question among

public health researchers. Compared to the measure of inequities, which are relatively easy to characterize, assessing health disparities or disparities in health conditions is not easy. Generally, the effort toward measuring health disparity has one main goal: to help inform transformative policies and provide justice. Health disparity assessment requires a sufficient understanding of health disparities and consistent measurement and variables that shape them. Consequently, several approaches have been used to measure health disparities. For example, disparities can be measured relative to a standard such as a Healthy People target [100].

Researchers have measured health disparities in absolute and relative terms for better comparisons across populations, geographic areas, and indicators [101,102]. Pairwise comparisons measure health disparity when the goal is to improve the health status of a particular group or population [103]. More specifically, the absolute measure of health disparity can determine the arithmetic difference between a group rate and a reference category. In contrast, a relative measure expresses the simple difference from the reference point as a percentage of the reference point [102]. Health disparities can also be measured relative to the total population represented by the domain of groups, using graphical displays over time and finding the ratio between the highest and lowest rates [104,105].

Nevertheless, these measures vary in terms of magnitude and the opposite direction in which these magnitudes can occur due to temporal changes [101]. In addition to measuring disparity in absolute or relative terms, several issues have been associated with measuring health disparities [102]. For example, researchers could choose to measure health disparity between groups in a pairwise manner or in a summary fashion, weighting groups according to size or in terms of favorable or adverse events [102]. The most common issue with measuring health disparity is the selection of a reference point from which to measure disparity. Selecting the best measure depends on the research question at hand, as this can affect the magnitude and direction of disparities measured at a point in time [102].

## 7. The Role of Geospatial and Machine Learning Techniques in Health Disparity

Since the global outbreak of COVID-19, several attempts have been made to investigate factors driving the variability of COVID-19 infection, mortality rates, and case-fatality rates/ratios at different geographic scales using different approaches. One well-known technique that has gained substantial popularity in public health and epidemiology research of infectious and noninfectious diseases is the use of geographic information systems (GIS). GIS has become an essential tool for revealing spatial disparity and distribution and understanding the change in disease patterns. Furthermore, to complement medical interventions, epidemiologists are keen on discovering the origin and diffusion rate of diseases using geospatial techniques for appropriate health intervention and preventing a future outbreak. Of the 40 selected studies in Table 2, 9 highlighted the application of machine and deep learning and artificial intelligence (ML/DL/AI), and 11 used spatial and aspatial techniques to study COVID-19, reinforcing the importance of innovative technologies in disease epidemiology.

### 7.1. GIS and Health Data Linkage

The integration of spatial and nonspatial data, as well as linking census data with electronic health data to create a database that can be queried for meaningful insight in supporting health policies, is possible through the application of GISs. Geographers, spatial epidemiologists, and other spatial scientists are at the forefront in marshaling the power of GISs using spatially encoded data such as mobility data to estimate and explain COVID-19 transmission patterns in the United States and beyond. Another contribution of the use of GISs to COVID-19 studies and health disparity investigation, in general, is its capacity to empower the linkage of health outcomes (morbidity or mortality) with socioeconomic, cultural, infrastructure, and environmental data at the aggregate level (different scales) to support the ecological investigation of the possible correlation between health disparity and socioecological factors (see, for example, [1]).

Undeniably, we are currently in the era of "big data," and the use of GISs has helped governmental and non-governmental agencies handle these enormous data that keep pouring near real-time. Based on the insights gained from manipulating and analyzing the available data, policymakers, and health agencies developed non-pharmaceutical interventions such as total or partial lockdown, travel bans for nonessential workers within a country, and international travel restrictions. What we have found is that the degree of implementation of various non-pharmaceutical measures and the level of the effective reproduction number ($R_t$) at the time of the intervention correspond to the suppression rate of COVID-19 in the United States, that is, the average number of people each infected person can spread the virus to [27].

Moving forward, geographically focusing on policies that will eliminate the most critical determinants of health, structural racism and poverty, is crucial to reducing health disparity among racial/ethnic groups. In this regard, persistent recognition of spatial data and scientists/geographers and their tools, such as GISs, would continue to be relevant in measuring and untangling health disparities.

### 7.2. Disease Epidemiology, Machine Learning, and Artificial Intelligence

Apart from GIS techniques, other sophisticated data analytics techniques such as machine learning (ML), deep learning (DL), and artificial intelligence (AI) are available to help mitigate several impacts of COVID-19. The field of machine learning is both broad and deep and is constantly evolving. ML is an innovative technique that has extensive applications in prediction. The adoption of machine learning in disease epidemiology provides more accurate and useful features than a traditional explicitly calculation-based method. Furthermore, ML algorithms can analyze the risk factors considering age, social habits, location, and environmental factors. ML is essentially used for predicting future events based on available data.

Interestingly, some GIS interfaces have built-in ML algorithms that can smoothly carry out some health-based analyses. For example, ArcGIS Pro, a proprietary GIS software by ESRI [17], has built-in ML techniques and DL algorithms, e.g., a forest-based classification and regression embedded in ArcGIS Pro based on the traditional random forest (RF) technique. It is important to note that RF is a supervised learning and classification algorithm that consists of many decision trees. In a technical term, the RF techniques use bagging and feature randomness when building each tree to create an uncorrelated forest of trees whose prediction is more accurate than any individual tree. In other words, RF makes a prediction based on random samples drawn from different chunks of data (forest) to obtain optimal outcomes compared to relying on a single sample (single tree). An excellent example of how ML and AI have been used in disease surveillance is the contact tracing of COVID-19 infection. In addition, using machine learning, data scientists interlink several forms of data such as environmental, population, and mobility data with health data to predict infection, death, or fatality rates [106–108].

Punn et al. [109] used a series of ML and DL techniques such as support vector regression (SVR), polynomial regression (PR), deep neural networks (DNNs), and recurrent neural networks (RNNs) to predict the total number of confirmed, recovered, and death cases worldwide. Similarly, Pinter et al. [110] used hybrid machine learning techniques, an adaptive network-based fuzzy inference system (ANFIS), and a multi-layered perceptron-imperialist competitive algorithm (i.e., MLP-ICA) to predict time series of individuals infected with COVID-19 and mortality rates in Hungary. The authors opine that ML techniques could be more useful in epidemiological modeling than the standard susceptible-infected-resistant (SIR)-based models.

Finally, coupled with geospatial techniques, ML, DL, and AI are powerful tools to identify health disparities, especially in the era of "big data." Hence, these techniques need to be constantly applied to identify populations at high risk and transmission patterns among communities and predict death rates, other possible abnormalities, and second-order impacts emanating from COVID-19 [111]. In addition, the accrued benefits of these innovative tech-

nologies would help public health experts avert unnecessary excesses in hospitalization and deaths from preventable diseases and asking the right questions [109–114].

## 8. Conclusions and Prospects

Health disparity is not a new challenge, nor is it going away easily without systematic and intersectional treatment. The COVID-19 pandemic provides a stage to reveal and re-examine the health disparities we have been facing in all aspects of public health, including health outcomes (infection, hospitalization, fatality, and various health complications), health care, and disease prevention (vaccine). Evidence of health disparities across different population groups manifested through all critical stages of the COVID-19 pandemic, from susceptibility to exposure, infection, and recovery or death.

The intersectionality approach can be used as a lens to understand the power structure and dynamics in a society that are central to the various social, political, and infrastructural issues leading to or sustaining health disparity. Treatment for health disparities calls for systematic tackling of these issues that place minority groups in disadvantaged positions for health and health services. Identifying the disadvantaged populations and communities and developing and implementing community-centered and place-based policies and practices to elevate the disadvantaged is core to reducing and eventually eliminating health disparity.

The continuous development of GIS and data science has greatly enabled our capabilities to promptly integrate and process unprecedently voluminous data to reveal health issues and disease patterns of particular challenges to specific populations or communities. As a result, community-centered and population-targeting health prevention and health service initiatives can be designed accordingly. We have seen some of these technologies being adopted to guide and optimize the distribution and administration of COVID-19 vaccines [115]. For example, a study in Flint and Genesee County, Michigan, investigated disparities in COVID-19 death using a GIS-based approach. Racial and location disparities detection resulted in public health interventions leading to prevention, testing, treatment, and vaccination rollout. With the world being interconnected and knitted together more than ever before, there is no room to leave any communities or population groups out to suffer from health disparity. As it sings in the song *We're All in This Together*, "Together together, come on let's do this right."

**Author Contributions:** Conceptualization, A.I.; investigation, A.I.; resources, A.I.; writing—original draft preparation, A.I., K.B., and Y.L.; writing—review and editing, A.I., K.B., and Y.L.; visualization, A.I.; and supervision, Y.L. All authors have read and agreed to the published version of the manuscript.

**Funding:** This research received no external funding.

**Institutional Review Board Statement:** Not applicable.

**Informed Consent Statement:** Not applicable.

**Data Availability Statement:** Data and literature were sourced from Google Scholar. The list of works is available in the References section.

**Acknowledgments:** Not applicable.

**Conflicts of Interest:** The authors declare no conflict of interest.

**Entry Link on the Encyclopedia Platform:** https://encyclopedia.pub/13863.

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
