# Peer review of "COVID-19: Evidenced Health Disparity"

_encyclopedia, doi:10.3390/encyclopedia1030057_

Round 1

Reviewer 1 Report

This is a well written review which encompasses the disease vulnerability of minority population in the United States. Authors have well discussed the topic with several sub-sections which are pre-dominantly focused on the cause and effects of disparity in general. I would like to make a few minor comments:

There are few places where the references are not provided, e.g.  add several references to subsection 3.1 in support of the statements.

A chart/table to demonstrate the level if disparity among the US states would make this review more interesting.

Author Response

Reviewer #1:

This is a well written review which encompasses the disease vulnerability of minority population in the United States. Authors have well discussed the topic with several sub-sections which are pre-dominantly focused on the cause and effects of disparity in general. I would like to make a few minor comments:

There are few places where the references are not provided, e.g.  add several references to subsection 3.1 in support of the statements.

A chart/table to demonstrate the level if disparity among the US states would make this review more interesting.

Response: Thank you for your feedback on our previous version. We have included additional references where appropriate. Section 3.1 now looks like this:

“Regarding disparity in COVID-19 outcomes, it is evident that the disease has exacerbated health inequality. The health impacts of COVID-19 are unevenly distributed, particularly for underrepresented racial and ethnic minorities and migrants [27–29]. Figure 3 shows the pattern and distribution of COVID-19 death based on the 2017 population density per square kilometer for some selected countries. It is worth emphasizing that the pandemic did not cause health disparity in that some minority ethnic groups died at a significantly higher proportion; it only shed more light on the longstanding health disparity in our country [30–33]. Existing data on infections, hospitalizations, and deaths reveal significant variation among and within regions and communities [27,28,34–36], prompting questions about health inequality and disparity; and have stirred curiosity to know which populations are at higher risk and why the risk is higher for one group than others [28]. The within-country health determinants may explain this variation among countries of the world. This type of question is what proponents of social, health and environmental justice have been pursuing before the outbreak of the COVID-19 pandemic.“

Reviewer 2 Report

Articles look like a news article. Please put more figures and stats to make your article good looking. Also, prepare a table from the literature to show the previously published related work and make a comparison to show the pros and cons of all previous studies. Also state how your study is better than previous ones.

Author Response

Articles look like a news article. Please put more figures and stats to make your article good looking. Also, prepare a table from the literature to show the previously published related work and make a comparison to show the pros and cons of all previous studies. Also state how your study is better than previous ones.

Response: Thank you for your feedback on the previous version of our manuscript. We have included some figures/tables. Based on your recommendation, we added Table 2, compiling some works done. Please, see attached file. Toward the end of page 2, right before Figure 2, we added this statement: “In addition, this paper contributes to the field of spatial epidemiology and public health by showing how data science and intersectionality framework can be fused to improve general/public health.”

Round 2

Reviewer 2 Report

accept